# Osteoarthritis of the Temporomandibular Joint: A Narrative Overview

**DOI:** 10.3390/medicina59010008

**Published:** 2022-12-20

**Authors:** Caroline Mélou, Pascal Pellen-Mussi, Sylvie Jeanne, Agnès Novella, Sylvie Tricot-Doleux, Dominique Chauvel-Lebret

**Affiliations:** 1CNRS, ISCR (Institut des Sciences Chimiques de Rennes), University Rennes, UMR 6226, 35000 Rennes, France; 2CHU Rennes, Pôle d’Odontologie, 35033 Rennes, France; 3UFR Odontologie, 35043 Rennes, France

**Keywords:** temporomandibular joint, osteoarthritis, review

## Abstract

*Background and Objectives*: This study reviewed the literature to summarize the current and recent knowledge of temporomandibular joint osteoarthritis (TMJOA). *Methods*: Through a literature review, this work summarizes many concepts related to TMJOA. *Results*: Although many signaling pathways have been investigated, the etiopathogenesis of TMJOA remains unclear. Some clinical signs are suggestive of TMJOA; however, diagnosis is mainly based on radiological findings. Treatment options include noninvasive, minimally invasive, and surgical techniques. Several study models have been used in TMJOA studies because there is no gold standard model. *Conclusion*: More research is needed to develop curative treatments for TMJOA, which could be tested with reliable in vitro models, and to explore tissue engineering to regenerate damaged temporomandibular joints.

## 1. Introduction

Osteoarthritis (OA) is a common chronic degenerative joint disease characterized by synovitis, cartilage destruction, and subchondral bone remodeling [1,2,3,4]. It occurs progressively and can lead to disability [5]. It can be either monoarticular or polyarticular [6]. The two types of OA include primary OA, which is idiopathic, and secondary OA, which is caused by trauma, iatrogenesis, and joint infection [7].

The temporomandibular joint (TMJ) is occasionally affected by chronic pathologies such as chondromatosis or OA [5,8,9]. OA can result in cortical bone erosion, flattening of joint compartments, osteophyte formation, subchondral sclerosis, and subcortical cyst formation [2,5,10]. Temporomandibular joint osteoarthritis (TMJOA) can lead to chronic pain and joint dysfunction including limited joint mobility and crepitus [5,7,11]. 

TMJOA is mainly treated symptomatically. There are several treatment strategies to relieve inflammation and prevent degradation of the joint complex. However, there is currently no effective strategy to repair and regenerate the damaged TMJ [12].

Although inflammation and remodeling of the subchondral bone occurs in TMJOA, the underlying pathogenesis is unclear [6,13], making it difficult to develop curative treatments.

The pathology of TMJOA has been poorly investigated [9], and it is difficult to compare the TMJ with other joints in the body because of certain unique characteristics. First, the TMJ is exposed to limited load-bearing forces, which makes it different from the other joints in the body. Second, the most superficial cellular layer is the fibrocartilage, which primarily contains type I collagen, and the deeper cellular layer contains type II collagen. This differentiates the TMJ from the other joints containing hyaline cartilage, which is made entirely of type II collagen [1]. Finally, the TMJ is a synovial joint that performs the most complicated movements that combine protrusion, retrusion, lateral excursion, and jaw opening and closing, making its movements three dimensional in the human body [13,14].

More research on TMJOA is needed to develop curative treatments [4]; therefore, it is necessary to review current knowledge in this field. The aim of this study was to summarize current literature on TMJOA, highlight possible prospects for curative treatments, and explore the possibilities offered by tissue engineering, through a narrative review.

## 2. Materials and Methods

The authors searched the MEDLINE online database for published articles on TMJOA using the following keywords: “Osteoarthritis” AND “Temporomandibular Joint.”

The inclusion criteria were as follows: articles on TMJOA, published between 2017 and 2020, and published in the English or French language. The exclusion criteria were as follows: articles on all temporomandibular joint disorders (TMD) without focusing on TMJOA, letter to editor, and articles published prior to 2017. The medical field is constantly evolving, which is why it was chosen to include only recent articles (after 2017).

The authors initially scanned the titles and abstracts and obtained 56 articles. A second selection was made after the full text articles were read, and 54 articles were shortlisted for this review. 

In addition, a second search of the MEDLINE online database was performed using the following key words: “Osteoarthritis” AND “Temporomandibular Joint” AND “Tissue Engineering.” Two articles, one published in 2016 and the other in 2018 were obtained. 

The search strategy is illustrated in Figure 1.

## 3. Results

A total of 56 articles comprising fifteen observational studies, six reviews, three case studies, one in vitro study, six combinations of in vitro and in vivo studies, twenty in vivo studies on animal models, three in vivo studies on human samples, one in vivo study on human and rat samples, and one systematic review and meta-analysis (Table 1) were obtained. 

### 3.1. Histology of the Temporomandibular Joint

Unlike other synovial joints with a hyaline cartilage, the articular surfaces of the TMJ (condylar process and articular surface of the temporal bone) are covered by fibrocartilage. In addition, the articular disc is composed of fibrocartilage [59]. 

Cartilages are hypocellular and avascular and contain only one type of well differentiated cells, the chondrocytes, which maintain the extracellular matrix (ECM) [60].

In the mandibular condylar cartilage, the ECM mainly comprises proteoglycans, which include biglycan, decorin, aggrecan, fibromodulin, and collagen [1,44,49,59].

The fibrocartilage is divided into three layers including the fibrous layer (i.e., the articular surface facing the disc), proliferation layer, and hypertrophic layer (adjacent to the sub-chondral bone) [13]. A previous study described a fourth layer between the proliferation and hypertrophic layers called the chondroblastic zone [59].

Among the collagens present in the ECM of the mandibular condylar cartilage, types I and III collagen are present in the superficial layer (i.e., the fibrous layer). Type II collagen is expressed in the deep layers, and type X collagen is present in the hypertrophic zone [1,53,59]. The immediate ECM surrounding the chondrocytes is the pericellular matrix, which in the mandibular condylar cartilage is mainly composed of type IV collagen, type VI collagen, and laminin [45,47].

Some authors have reported thickening of the fibrous layer of the mandibular condylar cartilage in TMJOA, which is a sign of inadequate repair, and the fibrous tissue is similar to granulation tissue in composition and appearance [38,53].

In contrast, the ECM of hyaline cartilage is primarily composed of collagen type II and proteoglycans, and is divided into superficial, middle, and deep zones [60,61]. Collagen fibrils in superficial regions are oriented parallel to the articular surface to reduce friction and shear stress. In contrast, the collagen fibrils in deep regions of the cartilage are oriented perpendicularly to the subchondral bone surface for anchorage [62]. The composition of the hyaline cartilage and fibrocartilage is shown in Figure 2.

### 3.2. Prevalence

TMJOA is a common condition, and its prevalence ranges from 8% to 60% [3,6,26,50]. It is difficult to obtain the exact prevalence because many asymptomatic patients have radiological signs of TMJOA [6,50]. TMJOA has a predilection for women [26,37] with a female-to-male ratio of more than 2:1 [27], which is similar to OA of other joints [63]. The prevalence of TMJOA increases with age [37]. Izawa et al. reported the prevalence of TMJOA by clinical and magnetic resonance imaging (MRI) examinations, estimating 25% prevalence in the 20–49 years age group and 70% in the 73–75 years age group [27]. Nevertheless, OA is thought to occur earlier in the TMJ than in other joints. Taleuan et al. reported that the average age of onset of TMJOA is 35 years, 10 years earlier than that of other joints [33], such as the knee. In addition, TMJOA affects not only adults but also adolescents [3]. Previous reports have described the early onset of TMJOA, especially during the pubertal phase [24]. 

### 3.3. Etiologies and Risk Factors

The etiopathogenesis of TMJOA remains unclear. OA is generally multifactorial [13,16,33] and the risk factors for TMJOA, including age, sex, genetics, infection/inflammation, and congenital and developmental abnormalities, are consistent with those for other joints [10]. Risk factors can be classified into local (trauma, parafunctions, and joint overload) and general factors (age, sex, and heredity) [33,56]. The mechanical stress exerted on the joint (trauma, parafunction, unstable occlusion, functional overloading, and increased joint friction) throughout life, which exceeds the normal adaptive capacity, appears to be of great importance in the development of TMJOA [13,18,32,33]. However, similar to the OA of other joints, innate immune responses are also important factors in TMJOA [19]. OA leads to non-specific activation of the innate immune system, which is similar to the chronic wound healing process [64].

In addition, the presence of disc displacement without reduction increases the risk of developing TMJOA [3,55]. Disc displacement causes stretching and inflammation of the retrodiscal tissue, often resulting in internal derangement of the TMJ and is a risk factor for OA development, with abnormal remodeling of the condyle and mandibular fossa [55].

Several authors have sought to elucidate the etiopathogenesis of TMJOA. Some studies have shown that progressive condylar resorption can occur after orthognathic surgery [32], while others have not found a relationship between skeletal patterns and TMJOA [20].

Izawa et al. reported the etiological role of the subchondral bone. They explained that a failure in subchondral bone remodeling can lead to chondrocyte damage and increased collagen degradation with the release of proteases and reduction of protease inhibitors, resulting in degradation of the ECM [27]. 

Dumbuya et al. also reported that patients with TMJ bone changes have a greater prevalence of connective tissue disorders because connective tissue disorders promote disc abnormalities (position and/or morphology) [26].

### 3.4. Pathogenesis

Although several factors and signaling pathways have been investigated, the pathogenesis of TMJOA is not yet well understood. The role of these factors is described as follows:

#### 3.4.1. Biglycan and Fibromodulin

Biglycan and fibromodulin are small leucine-rich proteoglycans present in the ECM that modulate signaling activity, including bone morphogenetic protein (BMP) signaling, of transforming growth factor (TGF) superfamily [49]. Nejad et al. reported that biglycan and fibromodulin bind to TGFβ1. They suggested that the absence of biglycan and fibromodulin would result in the accumulation of TGFβ1, which would increase chondrogenesis and ECM turnover and ultimately lead to an imbalance in ECM turnover, thus resulting in cartilage degradation [1]. Therefore, decreased expression of biglycan and fibromodulin may be biomarkers for early stage TMJOA, as suggested by Shirakura et al. [49].

#### 3.4.2. Hypoxia Inducible Factors and Vascular Endothelial Growth Factor 

Hypoxia-inducible factors (HIF) maintain normal chondrocyte function but activate destructive pathways if they are absent or upregulated. Cytokines can also deregulate HIF; HIF2α leads to cartilage destruction and HIF1α increases the expression of vascular endothelial growth factor (VEGF), which in turn leads to increased levels of MMP [4,40]. Angiogenesis during OA creates a hypoxic environment that activates HIF-1, which further upregulates VEGF expression. [65]. Angiogenesis appears to play an important role in the development of TMJOA. VEGF, a pro-angiogenic factor, may be involved in the pathogenesis of TMJOA [52] by binding to its receptor (VEGFR-2), resulting in the activation of extracellular signal-regulated kinase 1/2 (ERK1/2), which stimulates endothelial cells. In addition, ERK induces MMP. The VEGF contributes to the activation of the Notch pathway, which plays a role in vascular formation as well as proliferation, differentiation, survival, and apoptosis of cells [43,52]. The Notch pathway was investigated by Luo et al., who suggested that the pathway plays a role in TMJOA. In their study, TMJOA development was followed by Notch signaling activation in mice, and inhibition of the Notch signaling pathway tended to delay the progression of TMJOA. This has also been observed in patients with knee OA. Inhibition of the Notch signaling pathway enables a decrease in the target downstream of Notch signaling activation, including transcription factors hairy and enhancer of split (HES1, HES5) and MMP-13 [43]. Figure 3 summarizes the signaling pathways that seem to be involved in TMJOA.

#### 3.4.3. Estrogens

Changes in estrogen levels are considered as a cause of TMJOA in women [13], and previous estrogen studies have reported conflicting results [13,44]. However, high estrogen levels appear to be destructive to the condylar cartilage but protective to the subchondral bone [24,44]. Adequate estrogen concentrations are important for TMJ homeostasis [13]. ERα is an estrogen receptor, which, according to Wu et al., plays a role in mechanotransduction. They also reported that ERα can activate the ERK pathway, thus aggravating TMJOA [13]. Therefore, the ERK pathway may be involved in TMJOA [13,52].

Wu et al. reported that, due to an additive effect that involves the ERK pathway, a combination of estrogen deficiency and excessive mechanical stress on the condyle resulted in more severe TMJOA in a mouse model compared to that by estrogen deficiency or excessive mechanical stress alone [13].

#### 3.4.4. Other Factors

Studies have reported the role of the β-catenin signaling pathway. Inappropriate activity of this pathway can affect the integrity of the condylar cartilage [8,51].

The Indian Hedgehog signaling pathway has also been studied, and its activation appears to be induced in TMJOA [36,48].

Some authors have investigated autophagic activity in TMJOA because this activity is essential for chondrocyte survival and function due to the chondrocyte’s low rate of cellular turnover. Altered autophagic activity could contribute to several diseases. These authors reported that fibroblast growth factor (FGF) signaling activates the mechanistic target of rapamycin (mTOR), resulting in the suppression of autophagy via the Phosphoinositide 3-kinase/Protein kinase B (PI3K/Akt) pathway. They demonstrated that suppression of FGF signaling delayed the progression of TMJOA in mice, presumably by promoting autophagic activity [54].

Leptin appears to also play a role in TMJOA. In a study by Xiong et al., patients with TMJOA had significantly higher concentrations of leptin in their synovial fluid than that observed in healthy controls. The study also showed that leptin stimulates IL-6 expression, and leptin-induced IL-6 production was significantly reduced by the inhibition of JAK2/STAT3 (Janus kinase 2/Signal transducer and activator of transcription proteins 3), p38 MAPK (mitogen-activated protein kinases), or PI3K/Akt pathway [37].

The literature showed that IL-6 is highly expressed in the synovial fluid of patients with disc displacement. De Alcântara Camejo et al. investigated the expression of IL-6 in the articular disc of patients with disc displacement or OA. They found no significant differences between the disc displacement groups with and without reduction and between the groups with and without OA [55].

Yokota et al. also suggested that the ROCK/actin/MRTF axis promotes the fibrogenic activity of synoviocytes around the TMJ [34], and He et al. reported the role of the circadian Per2 gene in TMJOA [42].

Finally, Savage et al. investigated the presence of bacteria in condylar head samples of patients with TMJOA, but failed to identify meaningful bacterial growth [7].

### 3.5. Clinical Signs

The main clinical signs of TMJOA are joint noise (crepitus), loss of function with limited mouth opening, and pain [7,21,22]. Pain can be continuous and is aggravated during mastication [21]. Although these clinical signs are suggestive of TMJOA, the diagnosis is mainly radiological [10,32], with signs of cortical bone erosion, flattening of joint compartments, sclerosis, osteophyte formation, and subcortical cyst formation [5,10]. It is important to note that a direct relationship between radiological findings and TMJ symptoms may not always be present [10,16], and many patients with TMJOA are asymptomatic [18].

The disease is divided into several stages. Dumbuya et al. reported that bone resorption mainly occurs in the early stages, whereas osteophytes and subchondral cysts appear in more advanced stages [26].

Different classifications exist, including the Kellgren–Lawrence grading [16] and the Wilkes classification [55,58]. The Kellgren–Lawrence grading scale is used to classify OA of any joint, whereas the Wilkes classification is specific to TMD.

### 3.6. Diagnostic Tools

Accurate assessment of the TMJ using conventional radiography is difficult because of overlapping anatomical structures [20]. The imaging diagnosis of TMJOA is most reliably assessed by computed tomography (CT) [10]; however, CBCT appears to be the imaging tool of choice for TMJOA diagnosis for its cost effectiveness and accuracy at a reduced radiation dose [10,16,20,26]. Additionally, the available CBCT software provides detailed images to improve the identification of some alterations in the TMJ [20]. Liang et al. suggested that CBCT in combination with a software investigation protocol could detect trabecular changes in TMJOA, which could be beneficial because the trabecular bone might be a more sensitive indicator of early pathophysiological changes in OA [18].

When TMJOA is suspected, the following changes on the radiograph confirm the diagnosis: condylar flattening, osteophyte formation or loose bodies, erosion, deviation in form, subcortical or general sclerosis, subcortical cysts, hyperplasia or hypoplasia of the condylar head, and bony ankylosis [20]. Other criteria that can potentially help in the diagnosis of TMJOA have been investigated. For example, Lee et al. studied the increase in the horizontal condylar angle (the angle between the long axis of the mandibular condyle and the coronal plane perpendicular to the midsagittal plane in axial views) but concluded that it could not independently predict OA progression [19]. In contrast, the study by Zhang et al. demonstrated the importance of the osteochondral interface ossification in TMJOA diagnosis [46].

Other tools have been proposed for TMJOA evaluation such as positron emission tomography [4,15], hypoxia-detecting radioactive tracers, and single-photon emission CT [4]. Positron emission tomography is useful in diagnosing TMD, but also in assessing response to treatment [15].

In addition, TMJOA diagnosis might depend on biomarkers. Some pro-inflammatory and catabolic factors such as pro-inflammatory cytokines, which include tumor necrosis factor alpha (TNFα) and interleukin 1 beta (IL-1β) [4,44,56,66], matrix-degrading proteases such as matrix metalloproteinases (MMP, notably MMP-3, MMP-9, and MMP-13), and disintegrin and metalloproteinase with thrombospondin motifs (ADAMTS, notably ADAMTS-4 and ADAMTS-5) [1,8,41,44,48,55,66]. Therefore, these factors are potential markers for this condition. MMP-3 is an important biomarker of TMJOA and a predictor of its progression because it can cleave aggrecan, denature collagens, link proteins, and activate other MMP [41]. MMP are mainly expressed in OA joints, and few are expressed in healthy joints [41]. Sun et al. proposed an assay to measure the active form of MMP-3 in the human serum [67]. 

Some authors have attempted to highlight markers that can be used to predict the development of TMJOA. Zhang et al. showed that early growth response 1 (Egr1) reduction in peripheral blood leukocytes potentially indicates early-stage subchondral bone OA [57]. Reed et al. hypothesized that depletion of type VI collagen may be an early biomarker for the transition from early to end stage TMJOA [53].

### 3.7. Treatments

Treatment options for OA can be divided into noninvasive, minimally invasive, and surgical treatments [6,7,22]. However, rescue procedures may be considered in late-stage disease [6,33]. 

Noninvasive treatments, which need to be considered first, include pharmacotherapy (mainly non-steroidal anti-inflammatory drugs), physiotherapy, and occlusal splints [7,21,33,58].

Surgery is indicated only for severe and persistent pain and/or functional impairment when noninvasive and minimally invasive treatments such as arthroscopy, arthrocentesis, and intra-articular injections (hyaluronic acid and corticosteroids) fail to provide relief [7,21,33]. Surgical treatments include arthroplasty and condylectomy [7], whereas rescue procedures comprise total joint prosthetic replacement [33]. 

Arthrocentesis uses a joint lavage technique which removes inflammatory mediators and disrupts adhesions within the joint [22]. The effectiveness of arthrocentesis is controversial [6]; however, it improves pain and function [2,4,22]. A study by Nitzan et al. demonstrated good outcomes after arthrocentesis in patients who did not respond to noninvasive therapies and would have required surgical arthroplasty [2].

Arthroscopy has not been fully investigated. It includes biopsy, discopexy, (arthroscopic disc repositioning) synovectomy, and coagulation of tissues [68].

Treatment of TMJOA with intra-articular hyaluronic acid injections may be effective [4,22,25,41]; however, its use is controversial because the advantage over lavage is unclear, which is similar to steroid injections [4,22]. A meta-analysis performed by Chung et al. suggested that platelet-rich plasma injection is effective for pain reduction after arthrocentesis or arthroscopy in patients with TMJOA [58].

Two studies focused on the oral administration of glucosamine supplements [25,28]. The first study investigated the therapeutic effects of oral glucosamine as an adjunct to hyaluronic acid injection in patients with TMJOA. The authors concluded that patients who received oral glucosamine in addition to intra-articular injection of hyaluronic acid had a better outcome than that observed with patients who received oral placebo in addition to the injections [25]. The second was a systematic review of the efficacy of oral glucosamine compared with that of either a placebo or ibuprofen. The authors concluded that there was little evidence regarding the therapeutic effects of glucosamine on TMJOA [28].

In end-stage TMJOA, total reconstruction of the TMJ is required. Various modes including the use of autogenous bone tissue, distraction osteogenesis, and TMJ total prosthesis, stock, or custom-made can be used. TMJ total prosthesis can be used in end-stage TMJOA as well as in the final stage of other severe degenerative TMD such as ankylosis, severe condylar resorption, trauma, and cancer when conservative treatments have failed [31]. Since the 1990s, TMJ total prosthesis has been advocated instead of condylar prosthesis alone because condyle replacement alone can lead to resorption of the fossa [17]. TMJ total prosthesis has some advantages over grafting or other reconstructive methods, including the absence of a donor site, which reduces morbidity. However, it also has drawbacks such as implant fracture, intra-operative nerve damage, infection, etc. [31]. Ackland et al. described a 3D-printed prosthetic joint that seems promising, especially in cases with a pathological anatomy in which a commercial device cannot be used [31]. Boutault et al. highlighted that TMJ total prosthesis is often preferred to arthroplasty, but arthroplasty may be appropriate for elderly or fragile patients [21]. TMJ total prosthesis is schematically shown in Figure 4.

Finally, He et al. reported the potential of the use of low-intensity pulsed ultrasound for the early protection of cartilage [42], and AbuBakr et al. reported the potential effectiveness of low-level laser therapy [5], which is used to relieve pain in various musculoskeletal disorders [69]. This therapy is known for its stimulatory effects on tissue metabolism and its ability to modulate the inflammatory process after injury. It also causes better cellular oxygenation, the release of neurotransmitters associated with pain modulation, and the release of anti-inflammatory and endogenous mediators [5]. 

### 3.8. TMJOA in Adolescents

OA is generally an age-related condition that occurs in the weight-bearing joints; however, it can occur in adolescents and young adults [3]. 

The etiology of TMJOA in adolescents is not well understood. Female sex hormones and growth hormones play a role in female preponderance as well as the occurrence of TMJOA during the circumpubertal stages [24]. In addition, a study by Lei et al. suggested that TMJOA in young populations could be attributed to TMD such as disc displacement without reduction, and the prevalence of TMD in children and adolescents is increasing, which could partly explain the occurrence of TMJOA in adolescents [3].

TMJOA in adolescents can affect their growth as subchondral cortical bone formation only begins between 12 and 14 years of age [3]. Kang et al. reported that TMJOA could be associated with retarded dental development, mandibular backward positioning, and hyperdivergent facial profiles, but no association between TMJOA and skeletal maturation was reported. According to them, because a growing TMJ condylar surface lacks true articular cartilage, the adaptation process to overload differs from that of an adult condylar surface. Any process that destroys the condylar surface could interrupt normal condylar and mandibular development, resulting in a backward-positioned mandible and hyperdivergent facial profile. Retarded dental development may be influenced by abnormal mandibular growth [23]. Hong et al. reported that TMJOA can cause premature skeletal maturation and short predicted adult stature, particularly in female adolescents, due to increased stress and serum cortisol levels [24].

### 3.9. Models for the Study of TMJOA

There are many in vivo and in vitro models of OA. OA can be induced in vitro mainly by cytokine induction or load induction. These inductions can be performed on explants, chondrocyte cultures (in monolayer or three-dimensional cultures), or co-cultures [70].

Most studies included in this literature review used animal models [5,6,8,11,13,36,38,40,41,42,43,44,45,46,47,48,49,50,51,52,53,54,57]. The most commonly investigated animals were mice and rats [5,8,11,13,36,38,41,42,43,44,45,46,47,48,49,51,52,53,54,57]. One study was performed on rabbits [6]. Various animal models exist for the study of TMJOA using surgical (such as discectomy), mechanical (such as unilateral anterior cross-bite), chemical (such as intra-articular injections of monosodium iodoacetate or complete Freund’s adjuvant), and genetic techniques (such as Cre-recombination and knockout) [5,8,13,41,43,44,51]. The most commonly used techniques were discectomy [11,43,45,53], unilateral anterior cross-bite [36,38,44,46,54,57], intra-articular injections of monosodium iodoacetate [26,37], complete Freund’s adjuvant [41,48], and genetic techniques [8,49,51]. However, no gold standard animal model exists [42].

Monosodium iodoacetate is a metabolic inhibitor that disrupts the cellular aerobic glycolysis pathway and consequently induces cell death by inhibiting the activity of glyceraldehyde-3-phosphate dehydrogenase in chondrocytes. The resulting reduction in the number of chondrocytes and subsequent articular alterations were similar to those observed in human OA [71].

Complete Freund’s adjuvant acts by creating an immune response that leads to inflammation [72]. Studies that have conducted in vitro experiments have mainly used two-dimensional cultures from human [37,66] or murine [34,36,45,47,52] cells. 

### 3.10. Contribution of Tissue Engineering

To the best of our knowledge and based on this literature review, no tissue engineering strategies are used in clinical practice for TMJOA and, more broadly, for TMD. Nevertheless, Salash et al. underlined the importance of TMJ-related bioengineering research and detailed the indications and contraindications for the use of engineered tissues for the replacement of TMJ anatomic components [30]. The repair of disc thinning with tissue-engineered implants has been examined in minipigs, and the results seem promising [39].

Tissue engineering aims to regenerate tissues using three classic elements including scaffolds, growth factors, and cells [73]. 

Autologous chondrocyte implantation is a cell-based technique used to treat focal damage of the articular cartilage. The drawbacks of this procedure (such as donor-site morbidity and the need for a step of cell expansion that induces chondrocyte dedifferentiation, which could lead to fibrocartilage production) have encouraged the development of an alternative [74]. 

Mesenchymal stem cells (MSC) are attractive for tissue engineering because of their ability to regenerate damaged tissues [75]. They can be easily isolated from different tissue sources (bone marrow, synovium, adipose tissue, umbilical cord, dental pulp, etc.) [74,76]. The therapeutic effects of MSC are increasingly being attributed to paracrine secretion, particularly to nanovesicles called exosomes [77]. Investigation regarding the use of MSC exosomes as cell-free alternatives to MSC for tissue engineering, particularly, bone and cartilage tissue engineering, is increasing [73,78]. Some authors have considered the effects of these nanovesicles in TMJOA and reported that they alleviate TMJOA and promote osteochondral regeneration [77,78].

Regarding growth factors, BMP-2, VEGF, TGF-β1, insulin-like growth factor, and FGF have been investigated to develop tissue engineering strategies for the mandibular condyle [79].

## 4. Discussion

TMJOA is a fairly common condition that can occur earlier in life in comparison to OA of other joints, and can even affect adolescents [3]. During clinical examination, the presence of joint noise (crepitus), limited mouth opening, and pain can be an indication of TMJOA [7,21,22]; however, the diagnosis is currently made with radiological examination [10,32]. The imaging tool of choice for most TMJOA diagnosis is currently the CBCT [20]. On radiographs, TMJOA is mainly characterized by cortical bone erosion, flattening of joint compartments, sclerosis, osteophyte formation, and subcortical cysts [5,10]. Panoramic radiographs are widely used to view the TMJ but have poor reliability for detecting TMJOA [4,80]. Compared to CT/CBCT, MRI is considered to have lower reliability [2,10,80] but is the imaging tool of choice for visualizing soft tissues, altered disc position, and effusion [4]. Disc displacement is a risk factor for TMJOA and must be detected as early as possible [3,4,55].

The etiopathogenesis of TMJOA remains unclear, and OA is generally multifactorial [13,16,33]. Except for oral behaviors (parafunctions) [13], axis II issues of Diagnostic Criteria for Temporomandibular Disorders (DC/TMD) were not mentioned in the literature as possible etiologies. 

Anxiety was described in TMJOA and in general OA, but no direct evidence supported an increase in anxiety and depression in OA [81,82,83]. Nevertheless, stress and anxiety can aggravate parafunction [84].

TMJOA can result in complications such as ankylosis; therefore, there is a need for early treatment and diagnosis [33]. Furthermore, additional examinations may be required for patients with TMJOA. Abrahamsson et al. found that TMJOA was common in patients with hand OA, suggesting that TMJOA may be an indication of generalized OA [10]. Furthermore, Sonnesen et al. found that osseous upper cervical spine changes were more common in patients with TMJ changes compared to healthy individuals [16].

Several treatment options are available for TMJOA and are classified as noninvasive, minimally invasive, and surgical treatments [6,58]. TMJOA treatments, except surgical treatments that aim to reconstruct the TMJ, mainly aim to relieve inflammation and prevent cartilage and bone degradation. Currently, there is no effective treatment for rescuing the degraded cartilage [12,43]. Further research is required to develop curative treatments for TMJOA.

The lack of curative treatment is mainly due to the unclear pathogenesis of TMJOA [13]. Nevertheless, several pro-inflammatory and catabolic factors and signaling pathways have been investigated. Thus, the role of pro-inflammatory cytokines, TNFα and IL-1β [4,44,56,66], and matrix-degrading proteases, MMP-3, MMP-9, MMP-13, ADAMTS-4, and ADAMTS-5 [1,8,41,44,48,55,66] have been clearly defined. Estrogens are also important in the development of TMJOA [13]. Many other factors and signaling pathways such as HIF [4,40] and the signaling pathways ERK [13,27,35,51,52,66], Notch [43,52,54], β-catenin [8,51], Indian Hedgehog [36,48], FGF [34,51,54], p38 MAPK [27,37,66], and PI3K/Akt [37,51] have been investigated and require further research.

TMJ tissue engineering strategies have not yet reached the clinical trial stage and further research is warranted [39]; however, exosomes have been promising in this field [77]. The use of MSC exosomes associated with biomaterials could allow regeneration of damaged TMJ in the future. The mandible has a known potential for regeneration, and periosteum is the source of cells that are responsible for repair. The presence of the disc is necessary for the regeneration of the condyle. However, the reverse is not true; no disc regeneration can occur in the presence of an intact condyle [29].

TMJOA has not been extensively investigated [10] and requires further research, especially because the TMJ is a unique joint [1,13]. 

This literature review showed that TMJOA has mainly been studied in animal models. Only a few studies have performed in vitro experiments using two-dimensional cell cultures. Johnson et al. summarized in vitro models for the study of OA [70]. Several models are available; however, a gold standard has not yet been established. In addition, there is no existing reliable in vitro model for studying TMJOA. According to the “three Rs” concept, we must reduce the number of animals used, refine techniques to reduce pain and discomfort, and replace animal studies with alternatives [85]; therefore, the development of a reliable in vitro model for TMJOA is needed.

Despite the relevant observations in this review, it is important to note some limitations that potentially impacted the results, such as the large number of observational studies and the small number of systematic reviews and meta-analysis available. Additionally, the authors chose to include only recent articles, so articles prior to 2017 were excluded. This may have resulted in missing some relevant articles.

## 5. Conclusions

Further research is needed in the field of TMJOA to develop curative treatments. The development of a reliable in vitro model that would enable the testing of potentially curative treatments could be explored in the future. Tissue engineering, which aims to regenerate tissue, is an area worth exploring to identify a therapeutic approach that could be used to regenerate the damaged TMJ.

## Figures and Tables

**Figure 1 medicina-59-00008-f001:**
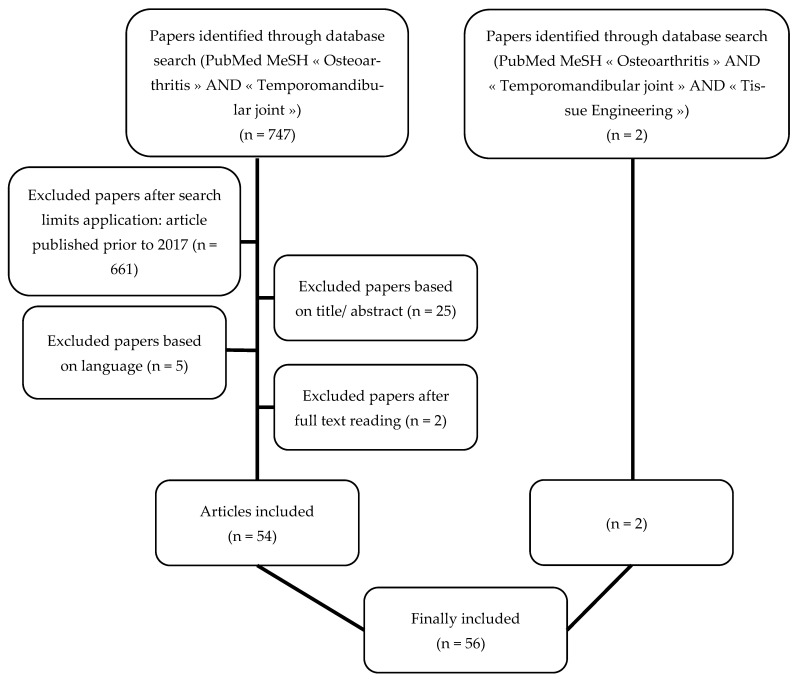
Flow chart of the search strategy: two searches were performed in the MEDLINE online database (PubMed). The authors initially scanned the titles and abstracts, and a second selection was made after reading the full text articles.

**Figure 2 medicina-59-00008-f002:**
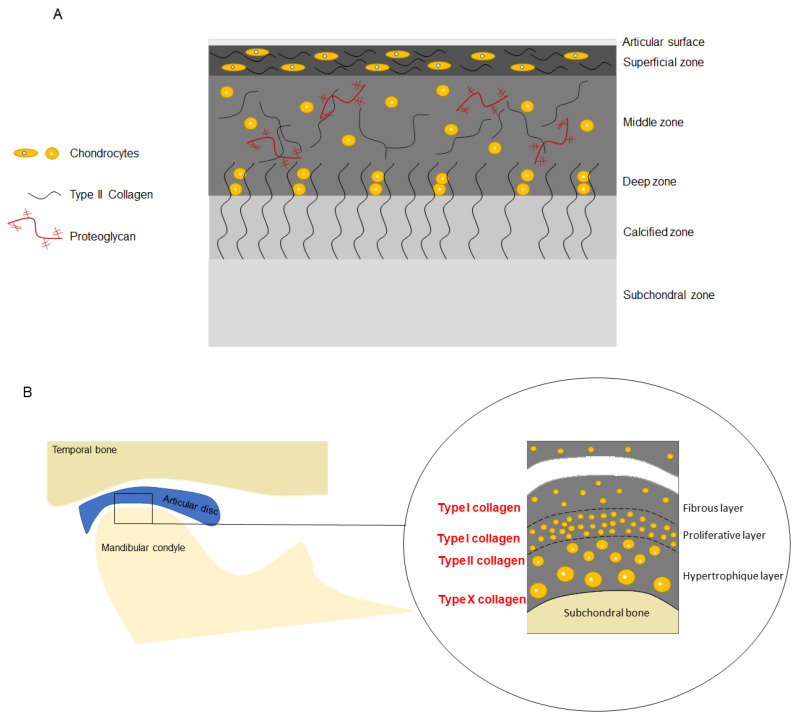
Composition of the hyaline cartilage (**A**) and fibrocartilage (**B**).

**Figure 3 medicina-59-00008-f003:**
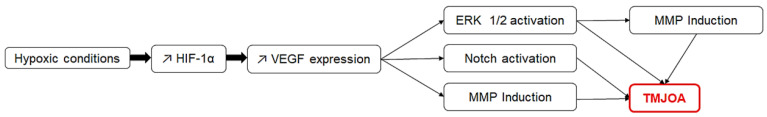
The signaling pathways that may be involved in TMJOA and their possible synergistic action.

**Figure 4 medicina-59-00008-f004:**
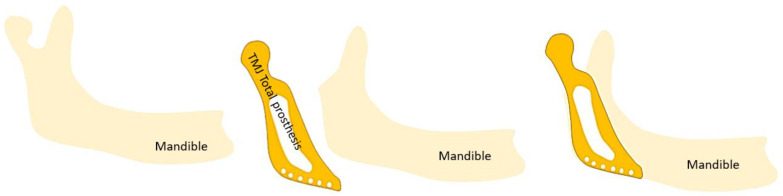
Schematic representation of a TMJ total prosthesis.

**Table 1 medicina-59-00008-t001:** Types of included papers.

Type of Article	Study First Author, Year
Observational Studies	- Abrahamsson, 2017 [10] - Suh, 2018 [15] - Sonnesen, 2017 [16] - Nitzan, 2017 [2] - Lei, 2017 [3] - Balon, 2019 [17] - Liang, 2017 [18] - Lee, 2019 [19] - Walewski, 2019 [20] - Boutault, 2018 [21] - Bergstrand, 2019 [22] - Kang, 2017 [23] - Hong, 2019 [24] - Cen, 2018 [25] - Dumbuya, 2020 [26]
Review	- Ghassemi Nejad, 2017 [1] - Sperry, 2019 [4] - Izawa, 2018 [27] - Melo, 2018 [28] - Roberts, 2018 [29] - Salash, 2016 [30]
Case study	- Ackland, 2018 [31] - Nojima, 2018 [32] - Taleuan, 2019 [33]
In vitro studies	- Yokota, 2017 [34]
Combination of in vitro and in vivo studies	- Alshenibr, 2017 [35] - Luo, 2019 [12] - Yang, 2019 [36] - Xiong, 2019 [37] - Zhang, 2017 [38] - Vapniarsky, 2018 [39]
In vivo studies on animal models	- Mino-Oka, 2017 [40] - Wu, 2019 [13] - Tolba, 2020 [41] - He, 2018 [42] - Luo, 2018 [43] - Ye, 2018 [44] - Yotsuya, 2020 [11] - AbuBakr, 2018 [5] - Memis, 2018 [6] - Hui, 2018 [8] - Yotsuya, 2019 [45] - Zhang, 2018 [46] - Chu, 2017 [47] - Xu, 2017 [48] - Shirakura, 2017 [49] - Chandrasekaran, 2017 [50] - Zhou, 2019 [51] - Dong, 2017 [52] - Reed, 2019 [53] - Wang, 2018 [54]
In vivo study on human samples	- de Alcântara Camejo, 2017 [55] - Savage, 2019 [7] - Monasterio, 2018 [56]
In vivo study on human samples and on rats	- Zhang, 2019 [57]
Systematic review and meta-analysis	- Chung, 2019 [58]

## Data Availability

Not applicable.

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
