# Peer review of "Osteoarthritis of the Temporomandibular Joint: A Narrative Overview"

_medicina, 2022, doi:10.3390/medicina59010008_

Round 1

Reviewer 1 Report

I congratulate the authors for a well-written and sensibly organised article.
It covers all important aspects of TMJ osteoarthritis pathology. Perhaps the only thing missing is clarification regarding the time frame of the literature search. In the title and methodology, it is necessary to explain that it is a narrative review article. Otherwise, I congratulate the authors for the effort they put into the exemplary preparation of the manuscript.

Author Response

Response to Reviewer 1 Comments

I congratulate the authors for a well-written and sensibly organised article.
It covers all important aspects of TMJ osteoarthritis pathology. Perhaps the only thing missing is clarification regarding the time frame of the literature search. In the title and methodology, it is necessary to explain that it is a narrative review article. Otherwise, I congratulate the authors for the effort they put into the exemplary preparation of the manuscript.

Response:

Dear Reviewer,

I thank you for the interest you have shown in our work.

Regarding the time frame, we have chosen to include only recent articles because the medical field is constantly evolving. A clarification has been made in the manuscript, at lines 82 and 83.

We have also explained that it is a narrative review article, in the title and at lines 73 and 74.

Reviewer 2 Report

The paper is good attempt in giving overview of TMJ osteoarthritis. The model discussed have good translational value.

Author Response

Response to Reviewer 2 Comments

The paper is good attempt in giving overview of TMJ osteoarthritis. The model discussed have good translational value.

Response:

Dear Reviewer,

I thank you for the interest you have shown in our work.

Reviewer 3 Report

Many thanks for the authors for this interesting paper. This paper reports an overview on osteoarthritis of TMJ: some modifications are required in order to proceed.

1)at line 47 the authors TMJ-OA to be a common pathology: this is not correct, occasionally is more suitable ; further the authors should report other disorders that can give symptoms and differential diagnosis with chronic osteoarthritis of the temporomandibular joint, such as chondromatosis; so please modify line 47 into

"..the temporomandibular joint (TMJ) is occasionally affected by chronic pathologies such as chondromatosis, osteoarthritis ....

please cite the following paper

Cascone P, Gennaro P, Gabriele G, Chisci G, Mitro V, De Caris F, Iannetti G. Temporomandibular synovial chondromatosis with numerous nodules. J Craniofac Surg. 2014 May;25(3):1114-5. doi: 10.1097/SCS.0000000000000667. PMID: 24739749.     2) figure 1: the caption should describe the flow chart, please modify   3)In the material and methods the authors selected only papers from 2017 to 2020: this is a little from international literature and limits the aim of the overview. Hovewer this limitation should be reported in the limit of the study in the discussion. Further, the authors should exclude letter to editor in the exclusion unless they have considered them in this study.

Author Response

Response to Reviewer 3 Comments

Many thanks for the authors for this interesting paper. This paper reports an overview on osteoarthritis of TMJ: some modifications are required in order to proceed.

1)at line 47 the authors TMJ-OA to be a common pathology: this is not correct, occasionally is more suitable ; further the authors should report other disorders that can give symptoms and differential diagnosis with chronic osteoarthritis of the temporomandibular joint, such as chondromatosis; so please modify line 47 into

"..the temporomandibular joint (TMJ) is occasionally affected by chronic pathologies such as chondromatosis, osteoarthritis ....

please cite the following paper

Cascone P, Gennaro P, Gabriele G, Chisci G, Mitro V, De Caris F, Iannetti G. Temporomandibular synovial chondromatosis with numerous nodules. J Craniofac Surg. 2014 May;25(3):1114-5. doi: 10.1097/SCS.0000000000000667. PMID: 24739749.  

Response 1:

Dear Reviewer,

I thank you for the interest you have shown in our work.

We have made this modification (at lines 48 and 49)

2) figure 1: the caption should describe the flow chart, please modify  

Response 2:

We added an explanation sentence in the caption, at lines 92-94.

3)In the material and methods the authors selected only papers from 2017 to 2020: this is a little from international literature and limits the aim of the overview. Hovewer this limitation should be reported in the limit of the study in the discussion. Further, the authors should exclude letter to editor in the exclusion unless they have considered them in this study.

Response 3:

We have mentioned this limitation in the discussion at lines 557-559 and we added letter to editor in the exclusion criteria (at line 82).

Round 2

Reviewer 3 Report

Article worth publication